# From Clean Labs to Noisy Lives: Real-World Stress Detection Using Spectrogram-Based Transformers on PPG Signals

Anice Jahanjoo, Yiting Wei, Soheil Khooyooz, Mostafa Haghi *Member, IEEE*, and
Nima TaheriNejad *Member, IEEE*

*Abstract*—Accurate detection of human stress levels is crucial for mental health monitoring and has wide-ranging applications in workplace wellness and healthcare. While Photoplethysmography (PPG) signals have been increasingly utilized to analyze physiological states, most existing studies are limited to controlled experimental settings. This paper addresses this gap by collecting PPG data from employees during real-world work conditions using a wearable device, thereby enhancing the validity and applicability of stress detection systems. We propose a novel stress classification method based on a multi-head attention transformer model, capturing both temporal and frequency-domain features in the PPG signal. We apply Short-Time Fourier Transform (STFT) to extract spectral representations of the PPG data. A transformer architecture is then employed to model complex dependencies and subtle variations in physiological signals via self-attention mechanisms and stacked encoder layers. Experimental evaluation demonstrates that our proposed method achieves a classification accuracy of 76.17% and an F1-Score of 76.42% in stress detection task, outperforming machine learning baselines and state-of-the-art methods. These findings highlight the effectiveness of transformer-based approaches in stress classification and reveal the substantial performance gap between laboratory-controlled results and real-world outcomes.

*Index Terms*—Attention mechanism, biomedical signal processing , photoplethysmography (PPG) , real-world data, stress detection

## I. INTRODUCTION

Workplace stress is becoming increasingly common, affecting employee health, productivity, and the economy [1]. Challenging tasks, rapid digital changes, and constant connectivity can lead to mental fatigue, poor decision-making, and lower performance, often resulting in increased absences and staff turnover. Digitalization plays a central role in this shift. While it fosters innovation and creates new job opportunities, about 40% of new employment in Organisation for Economic Cooperation and Development (OECD) countries occurs in sectors with high levels of digital activity [2]. This transformation also brings increased workloads, emotional exhaustion, technostress [3], digital stress, and telepressure, which refers to the pressure to respond instantly to digital communication [4]–[7]. These psychological pressures contribute to larger systemic challenges. Stress is the second most common work-related health problem in Europe after musculoskeletal disorders, which are often stress-induced themselves [8]. High work intensity and tight deadlines are leading causes, with 80% of managers acknowledging stress in their teams [9].

Generally, two types of stress can potentially contribute to the development of different diseases: chronic and acute. Persistent stress is connected to the onset of sudden events, and there was a tendency indicating that a higher level of acute stress is more strongly correlated with depression in individuals experiencing high chronic stress compared to those with low chronic stress [10]. In addition to depression, chronic stress exerts a notable impact on the immune system [11], increasing heart attacks and strokes, and eventually leading to the development of various illnesses [12].

Recent technological advances have enabled the development of sensors that monitor physiological states in real-time [13]. These sensors can be invasive, requiring implantation or attachment to specific body areas, which limits their practicality. Non-invasive alternatives, such as those embedded in wristbands, headbands, or rings, offer greater accessibility and user acceptance [14].

Photoplethysmography (PPG), as a non-invasive optical sensor, quantifies alterations in skin hue linked to changes in blood volume within subcutaneous vessels during the cardiac cycle [15]. It employs light pulses emitted by a source and captures the reflected signal using a photodetector [16].

In response to the growing importance of monitoring and controlling stress in real-world work environments, recent research has explored how continuous feedback from physiological sensors, such as PPG, can support improved well-being and performance [17], [18]. However, most existing studies are conducted in laboratory settings or rely on predefined scenarios within controlled environments [19]–[22]. While those studies offer important insights, they often miss the unpredictable and dynamic nature of real work environments. In this study, we collect PPG data directly from a real workplace, without setting up specific scenarios, asking participants to follow scripted tasks, or any intervention. This allows us to observe stress as it naturally evolves during the workday. Real-world signals often include significant noise and variability,

This work has been partially funded by the Vienna Science and Technology Fund (WWTF) [10.47379/ICT20034]; and Hector Stiftung [2304191].

Anice Jahanjoo is with TU Wien, Institute of Computer Technology, Gusshausstrasse 27-29, Vienna, Austria (e-mail:anice.jahanjoo@tuwien.ac.at).

Yiting Wei, Soheil Khooyooz, Mostafa Haghi, and Nima TaheriNejad are with the ECLECTX Team, Institute for Computer Engineering, Heidelberg University, Heidelberg, BW 69120, Germany (e-mail: firstname.lastname@ziti.uni-heidelberg.de)

which represents one of the most critical differences between real-world and lab-acquired data. This gap arises due to various factors such as motion artifacts, inconsistent sensor placement, or users wearing the wristband loosely, conditions that are typically controlled or absent in laboratory settings.

This paper aims to evaluate and improve the performance of detecting stress using PPG sensor data in real-world environments. To address the challenges of noisy, real-world physiological data, we first apply advanced signal processing techniques to remove noise and artifacts. We then use a hybrid deep learning model that combines Convolutional Neural Networks (CNNs) for local feature extraction, attention mechanisms to emphasize or prioritize informative segments of the input, and positional encoding to preserve temporal structure. This approach improves robustness and accuracy in stress detection under natural workplace conditions.

The remainder of this paper is organized as follows. Section II provides a detailed explanation of the dataset and data collection process. Section III presents our methodology, including preprocessing steps and the model architecture. Section Section IV details the experimental results, including performance evaluation, comparisons, and key findings. Finally, Section Section V concludes the paper with a summary of our contributions and outlines directions for future research.

## II. DATASET

In this study, we collected a PPG signal dataset in real-world office environments to investigate stress state recognition methods in natural workplace settings. A total of 25 healthy participants (18 females and 7 males) took part in the experiment, with an average age of 39 years, ranging from 24 to 55 years. Data collection spanned a continuous two-week period, during which participants wore the Empatica E4 wristband throughout their regular working hours. The device recorded their PPG signals in an unobtrusive manner, closely reflecting their actual working conditions. The complete data collection process is illustrated in Fig. 1.

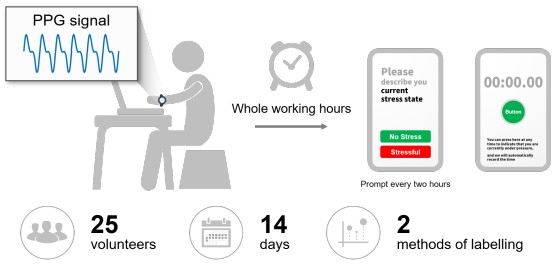

Fig. 1. Schematic diagram of dataset collection process

To obtain stress state labels, participants were instructed to report their current stress level (yes/no) via our custom-developed mobile application every two hours. This application was directly connected to the Empatica E4 wristband, receiving physiological data via Bluetooth and storing it locally on the mobile device as well as securely transmitting it to our dedicated server. Importantly, the system was designed

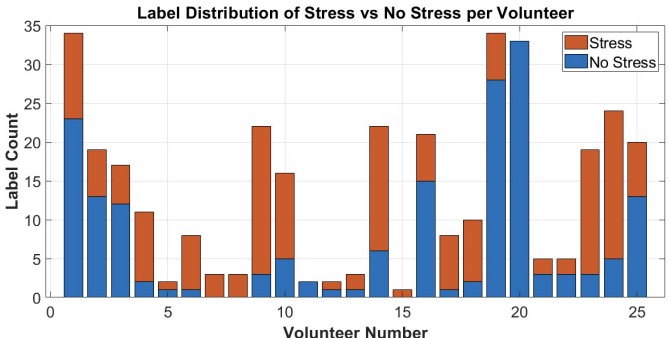

Fig. 2. Stress and non-stress labels per participant (scheduled and manual entries).

to function independently without relying on third-party cloud services, ensuring complete data privacy and control.

In addition to these scheduled prompts, the application also allowed participants to manually report stress at any moment by tapping a designated button. To ensure accurate temporal alignment with physiological signals, the system recorded the exact timestamp of both scheduled responses and manual entries directly on the server. These timestamps were critical in linking subjective stress reports with corresponding physiological data segments.

During data processing, for each stress label—whether scheduled or manually entered- a 10-minute segment of PPG data was extracted, spanning 5 minutes before and 5 minutes after the recorded timestamp. This segment was treated as the physiological signal window corresponding to the reported stress state. To provide an overview of labeling behavior across participants, we include Fig. 2, which illustrates the total number of stress reports recorded by each participant, separated into manually and prompted entries. The figure also distinguishes between stress and non-stress responses, offering insight into the distribution and frequency of reported stress states across the cohort.

Due to the real-world workplace setting without controlled lab conditions, participants were occasionally unable to respond to the scheduled prompts, whether due to being occupied, missing the notification, or forgetting to check the app. As a result, some participants contributed fewer labels than others, as shown in the figure.

Physiological signal acquisition was performed using the Empatica E4, a research-designed wristband shown in Fig. 3. This device's advantages include wearing comfort, lightweight, easy deployment, long battery life, on-device data storage, and secure data transmission, making it particularly suitable for long-term monitoring in naturalistic environments.

All participants signed informed consent forms prior to the study. The experimental protocol was approved by the Ethics Committee of the University of Vienna. To ensure participant privacy, all data were anonymized during both collection and storage. Each participant was assigned a random, non-reversible identifier, and no personally identifiable information, such as names, contact details, or device IDs, was collected.

## III. OUR PROPOSED METHOD

### A. Signal Filtering

We employed the noise detection method for PPG signals proposed by Khooyooz et al. in 2024 [23]. This approach used a machine learning model to automatically identify and exclude noisy segments from PPG recordings, ensuring that only clean signals are used for subsequent analysis. The detailed denoising process is as follows:

*1) Model Selection and Pretraining:* Among the various classifiers evaluated in the original work, the Extremely Randomized Tree (ERT) model demonstrated the highest classification performance, achieving F1-scores ranging from 89.3% to 99.4% in multi-class tasks. Therefore, ERT was selected for this study. To train the noise detection model, we utilized an external dataset containing PPG recordings with sensor specifications and sampling rates similar to those used in our study. Specifically, we used the PPG data from the publicly available dataset introduced by Gao et al. [24] and synthetically added noise to generate corresponding noisy segments.

Two types of features were extracted from each signal window for model training:

*a) Morphological Features:*

- Peak locations were identified using the *vital-sqi* open-access Python toolbox [25];
- The intervals between successive peaks were calculated;
- To standardize feature vector lengths, the following procedure was applied:
  - Sort all intervals in descending order;
  - Calculate the mean ($\mu$) and standard deviation ($\sigma$);
  - Set the final vector length as $l = \lfloor \mu \rfloor + \lfloor \sigma \rfloor$;
  - Apply zero-padding for shorter vectors or truncation for longer ones.

*b) Statistical Features:* The following statistical metrics were computed for each raw PPG signal segment:

- Mean, variance, kurtosis, skewness;
- Energy, entropy, maximum autocorrelation;
- Histogram mean, variance, and maximum value.

All features were normalized before feeding the model.

*2) Application to the Target Dataset:* Once the ERT model was trained, it was applied to our target dataset. We divided the PPG signals into fixed-length and non-overlapping windows.

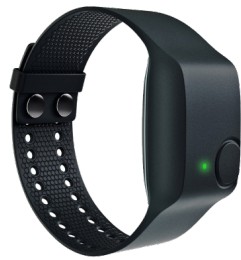

Fig. 3. Empatica E4 smart watch

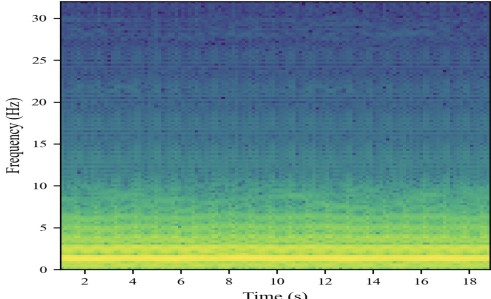

Fig. 4. Spectrogram representation of a 20-second noise-free PPG signal.

The window size was set to 1280 samples (equivalent to 20 seconds at 64 Hz). The same morphological and statistical features were extracted from each segment and fed into the trained ERT model, which predicted whether the segment was clean, corrupted by motion artifact, or affected by baseline wander.

*3) Noisy Segment Removal:* Finally, we removed all signal windows classified as either motion artifact or baseline wander from the dataset. Only segments classified as clean were retained for further analysis.

### B. Spectrogram-Based 2D Conversion of PPG Segments

Following signal filtering, each segment was treated as an individual instance, allowing us to isolate and analyze fixed-duration portions of the PPG signal. To capture both the temporal dynamics and frequency content within each window, we applied the Short-Time Fourier Transform (STFT) to produce corresponding spectrograms. As illustrated in Fig. 4, each spectrogram offers a visual representation of how signal frequency components evolve over time, formatted as a 2D array with dimensions 128 × 128, frequency bins along one axis and time bins along the other. This preprocessing strategy was designed to reshape the signal into a form suitable for CNN, enabling them to learn spatial patterns from the spectrogram data effectively. The following subsection outlines our method, which utilizes both CNN and transformer architecture to achieve accurate stress detection.

### C. Proposed Convolutional-Attention Network

We present a deep learning model designed to detect stress using PPG signals collected in real-world workplace settings. We derived spectrograms from the raw PPG signals, providing a time-frequency representation of cardiovascular dynamics. Stress is known to affect autonomic nervous system activity, which in turn influences heart rate variability, pulse amplitude, and rhythm changes that can manifest as distinct patterns in the time-frequency domain. Our model architecture combines convolutional layers with Transformer-based attention mechanisms [26], enabling the extraction of both fine-grained local features and long-range temporal dependencies from the spectrograms. The model architecture is illustrated in Fig. 5.

Our network begins with three convolutional blocks that process the input spectrograms ($1 \times 128 \times 128$), progressively

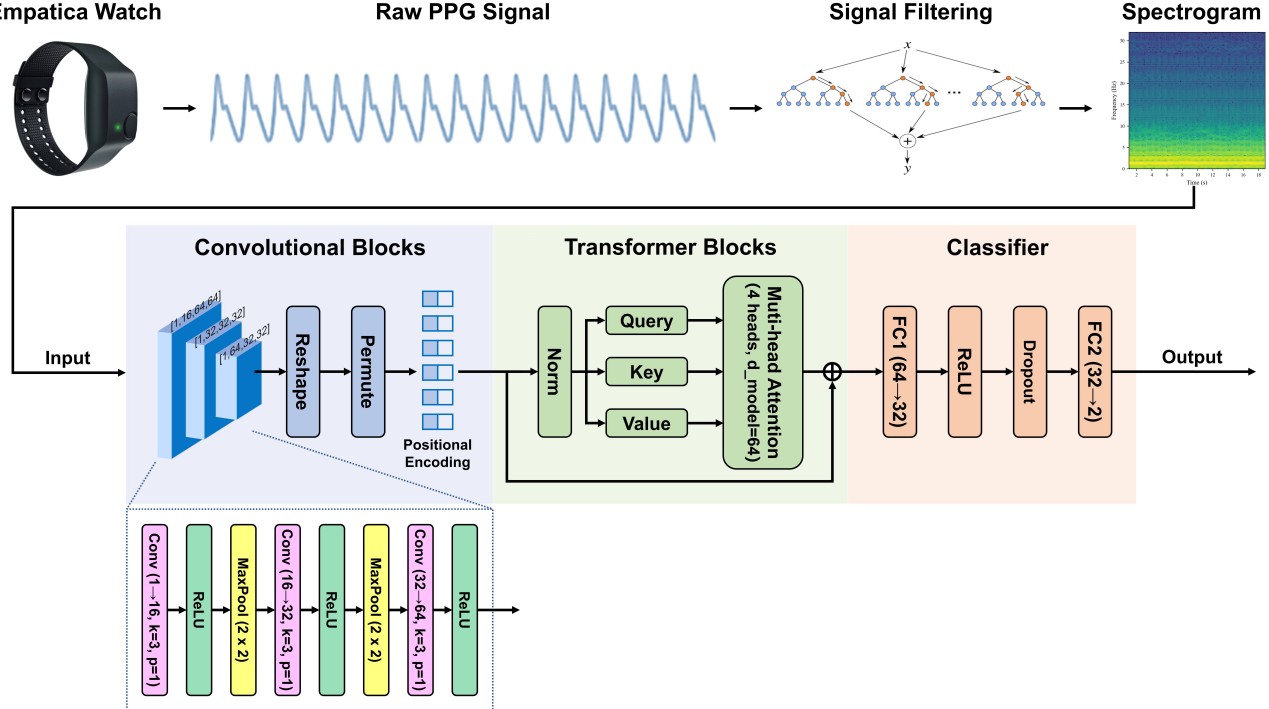

Fig. 5. Hybrid CNN-Transformer architecture for processing $128 \times 128$ spectrogram images.

learning higher-level features related to physiological stress markers. Early layers focus on lower-frequency rhythms, while deeper layers capture more complex spatiotemporal structures. We used Max-pooling to downsample the feature maps and reduce computational demands. To capture the temporal dynamics of stress, we reshaped the convolutional output into a sequence of tokens and passed it through a Transformer module. Sinusoidal positional encoding ensures that the model can differentiate between earlier and later time segments in the sequence, while multi-head self-attention models interact across the entire signal duration. We applied sinusoidal positional encoding, defined as:

$$\text{PE}(pos, 2i) = \sin\left(\frac{pos}{10000^{2i/d_{\text{model}}}}\right), \tag{1}$$

$$\text{PE}(pos, 2i+1) = \cos\left(\frac{pos}{10000^{2i/d_{\text{model}}}}\right), \tag{}$$

where $d_{\text{model}} = 64$ and $pos$ denotes the token position in the sequence.

The attention mechanism computes interactions across the full sequence using multi-head self-attention with 4 heads:

$$\text{Attention}(Q, K, V) = \text{softmax}\left(\frac{QK^{\top}}{\sqrt{d_k}}\right)V, \tag{2}$$

where $Q$, $K$, and $V$ are learned projections of the token embeddings.

After the attention layer, the sequence is aggregated using mean pooling and passed through two fully connected layers

$(64 \rightarrow 32 \rightarrow 2)$ to predict binary stress levels. This hybrid architecture leverages CNNs for extracting localized physiological features and Transformers for capturing the broader temporal context, offering a robust representation of stress patterns in real-world settings.

We implemented our model in PyTorch [27] and trained it on real-world data using the Adam optimizer. Training was conducted for 100 epochs on an NVIDIA RTX A2000 GPU with a batch size of 16, using mixed-precision to improve efficiency.

### D. Evaluation

To evaluate the performance of our binary stress classification model, we used four common metrics: accuracy, precision, recall, and F1-score. These are calculated as follows:

$$\text{Accuracy} = \frac{\text{TP} + \text{TN}}{\text{TP} + \text{FP} + \text{TN} + \text{FN}} \tag{3}$$

$$\text{Precision} = \frac{\text{TP}}{\text{TP} + \text{FP}} \tag{4}$$

$$\text{Recall} = \frac{\text{TP}}{\text{TP} + \text{FN}} \tag{5}$$

$$\text{F1-Score} = \frac{2 \cdot \text{Precision} \cdot \text{Recall}}{\text{Precision} + \text{Recall}} \tag{6}$$

These metrics offer a comprehensive assessment of model performance, capturing both correctness and the balance between false positives and false negatives in stress detection.

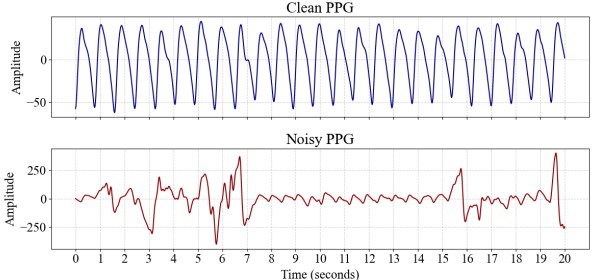

Fig. 6. Clean vs. noisy 20-second PPG signals highlighting real-world challenges.

## IV. RESULTS AND COMPARISION

We conducted stress detection on our real-world PPG dataset as a binary classification problem (stress vs. no stress). After applying the noise detection and filtering pipeline, we removed segments identified as noisy. Fig. 6 illustrates a model-predicted clean segment versus a noisy one, demonstrating the distinctions the noise detection model effectively identified and used to classify signal quality. Across all sessions in which the 25 subjects provided at least one self-reported label, we applied our noise detection and filtering pipeline. Following the removal of segments identified as noisy, an average of 886 ± 728 clean segments remained per subject, out of an original average of 997 ± 766 segments, resulting in a mean reduction of 13.002% ± 15.28%. Fig. 7 illustrates the distribution of total and clean segments across subjects. All segments were 20 seconds long, and from the remaining clean data, we selected 30 segments (equivalent to 10 minutes of recording) per subject for the classification task.

We converted each clean segment into a spectrogram representation, resulting in 10,620 distinct samples. We divided the dataset into 70% for training (7,434 samples) and 30% for testing (3,186 samples), while maintaining the class balances to ensure consistency. To improve the reliability of our evaluation and minimize overfitting, we performed 5-fold cross-validation within the training data. In this process, the training data was split into five equal parts; Each fold used 80% of the data for training and 20% for validation, rotating across subsets to provide a more generalized view of model performance.

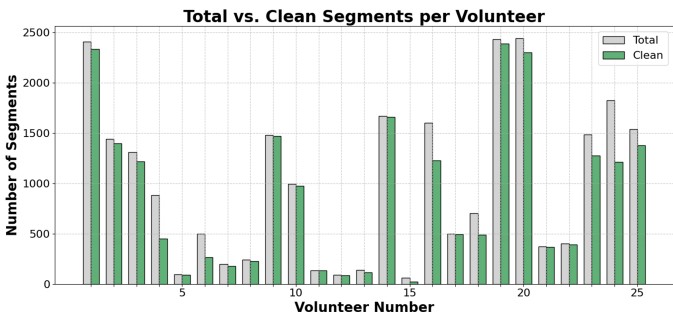

Fig. 7. Total vs. clean segments per subject after noise filtering.

TABLE I
PERFORMANCE COMPARISON OF PROPOSED MODEL WITH BASELINE CLASSIFIERS

| Model | Accuracy | Precision | Recall | F1-Score |
|---|---|---|---|---|
| Logistic Regression | 57.35% | 57.36% | 57.35% | 57.36% |
| Random Forest | 65.05% | 64.35% | 65.05% | 63.00% |
| KNN | 62.98% | 64.09% | 62.98% | 63.52% |
| SVM | 61.57% | 61.68% | 61.57% | 61.54% |
| XGBoost | 65.10% | 64.44% | 65.10% | 62.96% |
| CNN (Baseline) | 73.62% | 73.49% | 73.62% | 73.49% |
| Proposed Method | 76.17% | 75.67% | 77.19% | 76.42% |

Once the model was tuned, we evaluated it on the independent held-out test set to reflect real-world deployment conditions. Table I summarizes the performance of our proposed hybrid CNN-Transformer model using key metrics such as accuracy, precision, recall, and F1-score. In this table, we also compare our model's results with several commonly used baseline methods, including K-Nearest Neighbors (KNN), Support Vector Machine (SVM), logistic regression, XGBoost (XGB), and a CNN without attention modules. Except CNN, each baseline model was trained on a comprehensive feature set derived from the raw PPG signals. These features contained widely used time-domain descriptors, such as mean, standard deviation, variance, interquartile range, and Hjorth parameters, along with frequency-domain characteristics, including Fast Fourier Transform (FFT) statistics, spectral entropy, spectral centroid, and dominant frequency components. Additionally, we extracted band power measures using Welch's method across standard frequency ranges, as well as peak-based metrics, such as average peak height and peak intervals. This combination of features ensured that the traditional machine learning classifiers had access to both temporal and spectral dynamics of the signal for a fair comparison with our model. The configuration of the baseline models was as follows: KNN with $k = 9$, selected as the best from a range of 5 to 15 using Euclidean distance; SVM with a Radial Basis Function (RBF) kernel and default regularization parameter $C = 1.0$; logistic regression with L2 regularization using the `lbfgs` solver; XGB configured with a learning rate of 0.1, a maximum tree depth of 3, and 100 estimators; and a baseline CNN composed of three convolutional layers with 16, 32, and 64 filters respectively, each followed by ReLU activation and max-pooling, and concluding with a fully connected layer and a sigmoid activation function for binary classification. Our hybrid approach consistently outperforms all these baselines across every evaluation metric, demonstrating its capability to capture both local and global patterns in PPG signals for accurate stress detection.

To further assess the generalizability of our model and address subject-independent evaluation, we performed a leave-one-out evaluation. In this setup, each subject's data was used once as the test set, while the model was trained on the remaining participants. This evaluation provides a more realistic estimate of performance in real-world scenarios, where inter-individual variability is significant. Our model achieved an average accuracy of 74.58%, precision of 75.28%, recall

TABLE II
PERFORMANCE OF THE PROPOSED MODEL UNDER LEAVE-ONE–OUT
CROSS VALIDATION

| Metric | Accuracy | Precision | Recall | F1-Score |
|---|---|---|---|---|
| Mean | 74.58% | 75.28% | 75.34% | 74.31% |
| Standard Deviation | 13.78% | 15.48% | 14.24% | 14.55% |

of 75.34%, and F1-score of 74.31% across the participants. Table II presents the mean and standard deviation of the evaluation metrics computed across all subjects.

Compared to the 70/30 train-test split, the performance under LOO evaluation is slightly lower. This drop is primarily due to demographic variability, particularly age-related differences in PPG signal morphology. PPG signals from older participants, who are underrepresented in the dataset, often show characteristics that differ from those of the younger majority. When such individuals are used solely for testing, the model may encounter unfamiliar patterns, leading to reduced accuracy and higher standard deviation. These findings underline the importance of diverse and balanced data in developing robust physiological signal classification systems.

To compare our model's performance with existing approaches, we evaluated it against three existing approaches that were originally developed and evaluated on controlled laboratory datasets named Wearable Stress and Affect Detection (WESAD) [19]. Schmidt et al. [19] extracted time- and frequency-domain features from PPG signals collected in lab settings, achieving the highest stress detection accuracy of 85.83% using Linear Discriminant Analysis (LDA). Jahanjoo et al. [29] utilized PPG signals from the WESAD dataset. They applied novel denoising techniques, segmentation methods, and extracted a combination of seven key features from the PPG signals. Using a SVM classifier, they achieved a stress detection accuracy of 95.55%. Similarly, Heo et al. [28] reported a high accuracy of 95.07% also using WESAD PPG data. Their approach involved a robust preprocessing pipeline combining wavelet filtering and statistical techniques to reduce motion artifacts and baseline noise. Multiple peak detection methods were then used in parallel, and a voting mechanism to improve heart rate feature extraction. While these results are impressive, they reflect idealized conditions where signals are clean and variability is minimal. To assess the robustness and generalizability of these methods, we re-implemented them and evaluated their performance on our real-world dataset, which includes natural noise and variability despite preprocessing efforts. Under these realistic

conditions, all three methods showed a significant decrease in accuracy. Our proposed hybrid model, specifically designed to handle real-life signals, consistently outperformed these baselines when evaluated on our dataset. Although the absolute performance remains modest (76.17% accuracy), our results indicate the importance of designing models tailored to real-world data complexities and demonstrate the practical value of our approach for real-life stress detection.

## V. CONCLUSION

In this study, we presented a hybrid deep learning framework for robust stress detection using PPG signals collected in real-world workplace environments. We addressed the challenges of daily-life monitoring, where signals are subject to noise, motion artifacts, and non-stationarity. This study combines STFT with a multi-head attention Transformer model to precisely capture the time-frequency dependencies in physiological signals, achieving superior stress detection performance compared to existing methods in complex real-world environments.

To evaluate the effectiveness of our model, we compared its performance against classic machine learning classifiers, including KNN, SVM, logistic regression, XGB, Random Forest (RF), and a baseline CNN. We also re-implemented three state-of-the-art methods. Although these methods reported accuracies exceeding 85% in controlled lab settings, their performance dropped substantially on our dataset. In contrast, our proposed model achieved an accuracy of 76.18%, outperforming both classical baselines and the state-of-the-art lab-based approaches. To further validate our model's generalizability, we conducted an LOO evaluation. This approach better simulates real deployment scenarios by training on all but one subject and testing on the held-out individual. Our model maintained performance under this setting, achieving an average accuracy of 74.58% with a standard deviation of 13.78%. While slightly lower than the random-split evaluation, this result highlights the model's robustness in subject-independent settings and reveals the natural variation in physiological signals across individuals. In particular, differences in age distributions influenced the results.

While this result demonstrates the model's ability to handle noisy, non-stationary data and outperforms state-of-the-art, it also underscores a critical insight: the substantial gap between lab-based and real-world performance is not merely technical, but foundational. Future research will focus on deployment adaptability in real-world settings, particularly whether common wearing positions, such as the wrist, are suitable for acquiring high-quality physiological signals and accurate stress recognition. Further evaluation of the match between sensor types and placement sites will help improve the system's robustness and practical performance.

TABLE III
COMPARISON OF MODEL PERFORMANCE ON REAL-WORLD PPG DATA FOR
DETECTING STRESS

| Model | Accuracy | Precision | Recall | F1-Score |
|---|---|---|---|---|
| Schmitz et al. [19] | 57.61% | 57.64% | 57.61% | 57.61% |
| Heo et al. [28] | 66.73% | 68.96% | 64.63% | 66.73% |
| Jahanjoo et al. [29] | 70.03% | 71.62% | 67.35% | 69.04% |
| Proposed Method | **76.18**% | **75.67**% | **77.20**% | **76.43**% |

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
