# OpenReview forum: "From Clean Labs to Noisy Lives: Real-World Stress Detection Using Spectrogram-Based Transformers on PPG Signals"
_IEEE.org/EMBS/BHI/2025/Conference — BHI 2025_

### Official Review · Reviewer_MZXB · 2025-07-08
**Robustness, Visualization, Validation & Explainability of AI Concerns**

**Confidence:** 4
**Clarity Of Writing:** great
**Clinical Significance:** good
**Methodological Novelty:** fair
**Overall Rating:** 6
**Final Rating:** 7

**Experiments And Results:**

good

**Questions For The Authors:**

1. Does your ethics approval permit sharing this dataset with other researchers for further stress-detection work?

2. Could you discuss how the stress-detection features on commercial smartwatches (e.g., Apple Watch, Samsung Galaxy) compare in terms of robustness to your model’s predictions?

3. Although not mandatory, could you replace the block diagram in Fig. 5 with a more illustrative model schematic (e.g., a flow of data through layers), similar to the style used in 10.1109/TNSRE.2022.3230250?

4. Were the cross-validation folds strictly subject-independent (i.e., no participant’s data appeared in both training and test sets)? If not, could you clarify this to ensure that performance estimates are not inflated by data leakage?

5. If feasible, could you integrate an explainable-AI approach—such as applying Grad-CAM to the final CNN layer and visualizing multi-head attention weights—to demonstrate which parts of the signal the model focuses on when making stress predictions?

**Strengths:**

1. Collects a truly naturalistic PPG dataset (n = 25) without lab constraints.

2. Clear, concise writing makes the paper easy to follow.

3. Detailed, well-justified methodology and comprehensive comparisons to existing models.

**Summary Of The Paper:**

The study introduces a real-world stress-detection method using photoplethysmography (PPG) data from 25 adults wearing a wrist device during their normal workday. The paper proposes a hybrid CNN-Transformer model that achieves a 76.17 % accuracy on this dataset and benchmarks it against other state-of-the-art approaches.

**Weaknesses:**

1. The study includes only 25 participants' data, with no reporting of gender or other demographic breakdowns.

2. The hybrid CNN-Transformer architecture is presented without any analysis of which input features most strongly drive stress predictions, making it difficult to understand or trust the model’s decision process.

3. Performance results are given without standard deviations (or confidence intervals), despite using cross-validation. Including these measures is essential to assess the stability and reliability of the reported metrics.

---

### Official Review · Reviewer_EDCe · 2025-07-17
**Well structured paper and good approach, but require confirmation of possible overfitting**

**Confidence:** 4
**Clarity Of Writing:** great
**Clinical Significance:** good
**Methodological Novelty:** fair
**Overall Rating:** 4
**Final Rating:** 6

**Experiments And Results:**

fair

**Questions For The Authors:**

All my questions are at the weaknesses

**Strengths:**

The paper is well structured.
It demonstrates the application of stress detection using easy-to-use wearable devices in realistic scenarios, which is an important aspect for improving monitoring and health at work.
According to the authors, the results demonstrate the superior performance of the proposed model compared to other approaches in the literature.

**Summary Of The Paper:**

The present work shows a methodology to address the detection of stress in realistic scenarios by using the spectrogram of the PPG signal along with a novel approach based on CNNs and attention mechanisms.

**Weaknesses:**

Some comments should be addressed:
- Fig. 5 is a little confusing when it comes to locating the ReLU and output. It seems that this information comes from the previous layer rather than the actual convolution layer. I recommend placing this information after each block rather than before.

- After segment removal, how many samples remained in total? Indicate this with the mean ± standard deviation of the samples per subject and the percentage reduction from the original number of samples.

- According to the manuscript, it is unclear whether a subject-wise split was ensured in the training/testing division and cross-validation to avoid data contamination and train-test overlap of signals from the same subject, which could lead to overfitting.

- It may be useful to show a noisy segment that has been detected versus a clean one.

- Additionally, a flowchart of the complete processing-classification process could improve the manuscript.

---

### Official Review · Reviewer_zVuR · 2025-07-17
**Solid Paper with good experimental results**

**Confidence:** 4
**Clarity Of Writing:** excellent
**Clinical Significance:** great
**Methodological Novelty:** great
**Overall Rating:** 7

**Experiments And Results:**

great

**Questions For The Authors:**

- Regarding the labeling protocol: Could you provide more detail on the label collection process, such as the participant compliance rate for the bi-hourly prompts and the distribution of "stress" vs. "no stress" labels? How did you consider the potential temporal mismatch between the moment a stressor occurs and when the participant reports it?

- Regarding the denoising step: What fraction of the total collected data segments were classified as noisy and removed by the ERT model? Could this filtering process be inadvertently removing challenging-but-valid instances of stress that co-occur with motion, thereby simplifying the problem space?

- The Transformer module operates on 20-second windows. Given that workplace stress can be both acute and chronic, have you considered architectural changes to model longer-term dependencies, perhaps by processing sequences of windows to capture how stress states evolve over several minutes?

**Strengths:**

- Real-World Problem Focus: The paper's most significant strength is its direct confrontation with the "lab-to-real-world" problem. By collecting and analyzing data in a naturalistic workplace setting, the study provides a much-needed dose of realism to the field of wearable-based stress detection.

- Insightful Evaluation: The experimental design is excellent. Comparing against not only standard ML baselines but also re-implementing and testing high-performing lab-based methods on their own dataset is a powerful approach. This directly and quantitatively supports their central thesis that real-world data presents unique challenges not captured in controlled settings.

- Novel and Well-Motivated Architecture: The proposed hybrid CNN-Transformer model is a modern and appropriate choice. It logically combines the strengths of CNNs for extracting local patterns from the spectrograms with the power of Transformers to model complex temporal relationships, which is ideal for physiological signals.

- Clarity and Reproducibility: The paper is well-written, with a clear narrative and logical flow. The methodology, from data collection to model architecture, is detailed sufficiently to aid understanding

**Summary Of The Paper:**

The authors propose a multi-stage process for detecting stress from real-world PPG signals. First, data was collected from 25 employees over two weeks using an Empatica E4 wristband, with binary stress labels (yes/no) self-reported via a custom app . The collected PPG signals are segmented, and a pre-trained machine learning model is used to identify and remove noisy segments . Clean 20-second segments are then converted into 128x128 spectrograms using the Short-Time Fourier Transform (STFT). Finally, a hybrid deep learning model, combining convolutional layers (for local feature extraction) and a Transformer block (for modeling long-range dependencies), classifies these spectrograms as "stress" or "no stress".

The proposed hybrid model achieved a 76.17% accuracy and 76.42% F1-Score. This performance was superior to all classical ML and baseline CNN models. Crucially, the lab-based SOTA methods showed a substantial drop in accuracy when applied to the real-world data (performing between 57.61% and 70.03%), while the proposed model outperformed them all in this realistic setting.

The primary contributions are the development of a novel spectrogram-transformer architecture for real-world PPG-based stress detection and the empirical demonstration of the performance gap between models developed in controlled labs versus those tested in noisy, real-life environments.

**Weaknesses:**

- Opacity of the Denoising Step: The study uses a pre-trained ERT model to filter the data. The performance of the final stress detection model is therefore dependent on the accuracy of this initial filtering step. It is also unclear what proportion of the originally collected data was discarded as "noisy." If a large amount of data was removed, the model may have been trained on an overly "cleaned" subset of the real-world data, potentially underestimating the true difficulty of the problem.

- Modest Absolute Performance: While outperforming all baselines is a strong result, a final F1-score of ~76% for a binary classification task indicates that the model is still far from perfect. This underscores that real-world stress detection remains an exceptionally difficult open problem.

---

### Official Review · Reviewer_D7ny · 2025-07-18
**Interesting dataset and problem, but needs clarification on dataset construction and train test split**

**Confidence:** 4
**Clarity Of Writing:** fair
**Clinical Significance:** good
**Methodological Novelty:** fair
**Overall Rating:** 4
**Final Rating:** 6

**Experiments And Results:**

good

**Questions For The Authors:**

Was the tokenization step as simple as pivoting the array? Or were there more steps included? Currently the only description is "we reshaped the convolutional output into a sequence of tokens."

**Strengths:**

The collected dataset is large, providing ample positive and negative support.

This paper presents a useful real-world use of wearables signals.

Classification performance is solid, considering the paper only utilizes the PPG sensor, and none of the other sensors from the device.

**Summary Of The Paper:**

In this paper, wrist-worn wearable sensors are used to predict acute stress events in free-living work conditions. Empatica E4 wristbands were deployed for 14 days for 25 volunteers, who annotated whether they were stressed or not at 2 hour increments, or as acute events occurred. The PPG signal was used to make predictions on whether or not a person was stressed at a given time. A feature engineering approach is compared with a CNN + Attention model, where the CNN + Attention model consistently outperforms the other models. Further, the final model outperforms adapted models from literature whose performance drops significantly when transitioning from laboratory conditions to free-living conditions.

**Weaknesses:**

Some of the verbiage needs to be streamlined, for example "This device advantages such as wearing comfort, lightweight and easy deployment..." in section II.

The method of going from 14 days of data for 25 participants into a dataset of 10,620 samples is unclear. That means that on average, each individual only has 2.36 hours in the final dataset (if each sample is 20 seconds). Is it only the 10 minute segments surrounding each survey that end up in the final dataset? (section IV).

The described train-test-split is vague. The paper needs to clarify if any actions were taken to minimize data leakage. Was it ensured that the windows from each 10-minute segment would all be put in either the train or test set? If adjacent windows from the same stress event are divided into the train and test set, there is strong reason to believe this is not measuring generalization. The best case scenario is that the train test split occurred at the participant level. Otherwise, temporal separation should occur so that the model doesn't learn proxies for each individual stress event.

---

### Official Review · Reviewer_35Lt · 2025-07-18
**Translational work with real-world experiments but doubtful evaluation and unclear dataset measures**

**Confidence:** 5
**Clarity Of Writing:** great
**Clinical Significance:** good
**Methodological Novelty:** good
**Overall Rating:** 5
**Final Rating:** 6

**Experiments And Results:**

fair

**Questions For The Authors:**

As above.

**Strengths:**

The dataset is collected in real-world and represents a longer monitoring period. The paper is thoughtful about the quality of the PPG data instead of feeding the data into a ML model directly. The paper compares the proposed method with multiple benchmarks.

**Summary Of The Paper:**

The paper evaluated PPG-based stress detection in real-world workplace environment. PPG data is collected from 25 healthy volunteers during their normal work for two weeks period. The paper developed a CNN-Transformer-based modeling to classify stress/non-stress from the filtered PPG signal. The developed modeling outperformed conventional features-ML modeling as well as other baselines developed in dataset obtained under controlled conditions.

**Weaknesses:**

Though the paper's focus on real-world and long monitoring duration is commendable, some key aspects of the work are not presented which hampers evaluation and appreciation for the work. In particular,
- it is not clear what are the class distributions of the reported stress and how does it differ across participants.
- the paper seems to have used random 70/30 split for evaluation (and single split). This split combined with extracting multiple segments from a labeling window could heavily bias the results. Co-located segments which share labels and share confounders could be easily present in both the training and test set. Could it be possible to report results for leave one participant out evaluation and 5-fold random crossvalidation in the outer loop also.

Minor points:
1. In response to the growing importance of understanding stress in real-world work environments, a growing body of
research explores self-optimization using physiological sensors to support well-being and performance ==> Not clear what self-optimization means in this context.

2. To ensure participant privacy, all data were rigorously anonymized during ==> please be specific about how it was rigorously anonymized

3. For model pretraining, we selected a dataset with the exact specifications used in previous work as our
pretraining dataset ==> please provide citation or more details about this dataset.

4. III.A Signal Filtering section - It will be better to report what percentage of data, on average, was retained/filtered.